# Unifying Vocabulary of Large Language Model with Statistical Token-level Alignment

## Abstract

Large Language Models (LLMs) achieve great success across many general tasks, but the mismatch among different vocabularies hinders further applications like token-level distillation and inference with various models. To align the vocabularies of LLMs, we propose a simple yet effective method named **UnifyVocab** to replace the vocabulary of an LLM at a limited cost. A new vocabulary alignment method is devised first to align the source vocabulary to the target one. We then rearrange the corresponding parameters like embeddings, and progressively fine-tune the model. Experimental results on models across multiple parameter scales demonstrate the effectiveness and generalization of UnifyVocab, which costs as few as 10B tokens to recover 98.02% performance of the vanilla models on average. We further find that unifying the vocabularies significantly facilitates the token-level distillation which remarkably boosts (+4.4%) the model with only 235M tokens. Moreover, our method provides a better initialization of multilingual vocabulary for LLMs to adapt to new languages.

## 1 Introduction

Large language models like LLaMA, GPT-4, and Qwen (Touvron et al., 2023b; OpenAI, 2023; Qwen, 2024) show impressive general abilities. These models have specific strengths and weaknesses, which arise from their pre-training corpus and method. However, the mismatch among their vocabularies impedes the deep knowledge transfer between these models like token-level distillation and ensemble. Thus, it is important to unify the vocabulary of the large language model at a low cost.

The vocabulary of the language model is kept unchanged after pre-training unless adapted to a new language. It is common to append new tokens to improve the effectiveness of encoding on a new language (Tran, 2020; Wang et al., 2020; Chau et al., 2020; Minixhofer et al., 2022; Cui et al., 2023; Liu et al., 2024).

In this paper, we introduce a method called **UnifyVocab** to replace the vocabulary of large language models from a view of token-token co-occurrences. As the general process to train an LLM, the pre-training corpus is first tokenized into token IDs, and then input into the model. Given the same pre-training corpus, different tokenizers result in various sequences of token IDs, while the semantic and syntactic information is preserved in the token-token co-occurrence. Therefore, UnifyVocab strives to align the token IDs from the original vocabulary and the target ones based on the global token-token co-occurrence matrix (Pennington et al., 2014). We further propose a metric to evaluate the performance of the token-token alignment matrix. The new embedding and language modeling head of LLMs ("$lm\_head$" in the transformers (Wolf, 2019)) are initialized from the re-arranged parameters using the learned alignment matrix. Further adaptation process for the new vocabulary is divided into a progressive two-stage procedure to improve the stability of convergence.

Given a target vocabulary for substitution, results on models across different scales show that as few as 10B tokens are needed for our method to recover 98.02% performance of vanilla models on average. The training process of UnifyVocab is 1.92x faster than the best baseline method. Unifying vocabulary further facilitates the token-level distillation between models, which is 4.4% better than the sentence-level distillation on the same corpus. In addition, the model trained on the English corpus obtains a good initialization for the multilingual vocabulary, decreasing the perplexity from $2.9e^5$ to $2e^2$, and could adapt to new languages with only 4B tokens using UnifyVocab.

Figure 1: Illustration of UnifyVocab to align the token IDs from different vocabularies. We train token representations on the tokenized corpus, and align token IDs by the cosine similarity. It is noted that the IDs of tokens belonging to both vocabularies are directly replaced without alignment.

To sum up, our contributions are as follows:

- We propose a general method to align token IDs between two vocabularies and replace the vocabulary of the large language model from the token-token co-occurrence view, which costs as few as 10B tokens in the new vocabulary adaptation.
- We introduce a metric to evaluate the performance of token-level alignment, which is found proportional to the initial loss of pre-training.
- Experimental results show that our method promotes deep knowledge transfer between models like token-level distillation, and even the cross-lingual knowledge transfer among multiple languages.

## 2 UNIFYVOCAB

### 2.1 VOCABULARY ALIGNMENT

As shown in Figure 1, there are three steps in UnifyVocab to align two vocabularies of language models from the token-token co-occurrence information. We denote the source tokenizer as $\text{Tokenizer}_s$, which has $\mathcal{V}_s$ tokens, and the target tokenizer as $\text{Tokenizer}_t$ with $\mathcal{V}_t$ tokens, correspondingly.

**Step 1: Tokenization** The comprehensiveness of the pre-training corpus is important to obtain a well-trained token representation. An unbalanced corpus makes it hard to train the representation of tokens in the tail of vocabulary. Thus, the corpus used in this work is empirically composed of multilingual corpus CulturaX[40%] (Nguyen et al., 2023), code corpus The Stack[30%] (Kocetkov et al., 2023), and math corpus Proof-Pile-2[30%] (Azerbayev et al., 2024). We tokenize the mixed corpus using various tokenizers of different LLMs, and obtain multiple sequences of token IDs for the same corpus. The default token amount of corpus used in this step is 1B, which is investigated in Appendix B.1.

**Step 2: Token Representation Learning** We adopt GloVe (Pennington et al., 2014) to train the representation of the tokens from Step 1. The main reason is that GloVe considers more global statistical information than those slide window methods like CBOW and fastText (Mikolov et al., 2013a;b; Bojanowski et al., 2017). The details of training settings for GloVe vectors are reported in Appendix A.

**Step 3: Token Alignment** Based on the assumption that token representations capture the semantic information in the token, we align token IDs using the pair-wise cosine similarity of learned token representations. It should be noted that the ID of tokens belonging to both vocabularies are directly

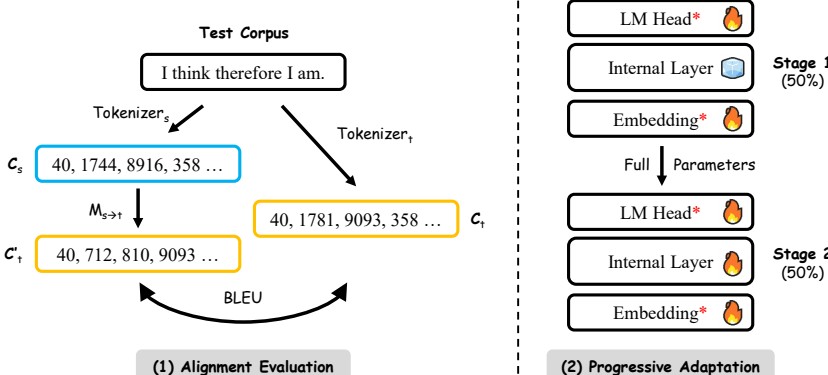

Figure 2: (1) We choose BLEU to evaluate the performance of alignment matrix $M_{s \to t}$ (2) The embedding and lm_head are tuned at the first half part of the tuning process, which follows the full parameter tuning. * indicates the parameter of each target token is initialized from the one of the most similar source token by alignment matrix $M_{s \to t}$.

replaced without the need to align. We denote the token-token alignment matrix $M_{s \to t}$, which maps the token id from the source vocabulary to the one with the highest cosine similarity from the target vocabulary.

## 2.2 ALIGNMENT EVALUATION

Figure 2(1) illustrates our metric to evaluate the performance of alignment matrix $M_{s \to t}$. We first tokenize the test corpus $\mathcal{C}$ using different tokenizers, which results in $\mathcal{C}_s$ and $\mathcal{C}_t$. The token ID corpus $\mathcal{C}_s$ from the source tokenizer is converted by the alignment matrix $M_{s \to t}$, and comes to the corpus $\mathcal{C}'_t$. The higher BLEU score between $\mathcal{C}'_t$ and the corpus $\mathcal{C}_t$ from the Tokenizer$_t$, the better alignment matrix $M_{s \to t}$ is. The other two metrics, BLEU-1 and BertScore, to evaluate the performance of alignment matrix are investigated in the Appendnix B.4.

## 2.3 PROGRESSIVE ADAPTATION

Given the alignment matrix $M_{s \to t}$, the parameters of each token in the target vocabulary are initialized from the ones of the most similar source token. We find that these re-arranged embedding and lm_head provide a good initialization for the new model (Section 3.2 and 4.2). Figure 2(2) illustrates the two stages designed for a LLM to adapt to the new vocabulary. The re-arranged embedding and lm_head are tuned first to avoid loss spike and improve the stability during tuning (Figure 5(c)). The other parameters of internal layer are further tuned together in the last half part. We acknowledge that a better designed adaptation method can bring a higher performance, which can be investigated in the future.

# 3 EXPERIMENTS

## 3.1 EXPERIMENTS SETTINGS

**Large Language models** We adopt the fully open-source language model series Pythia (Biderman et al., 2023) as base models in this work. It is noted that this work does not intend to achieve the state-of-the-art performance of large language models but rather investigate an effective method to replace the tokenizer. To achieve token-level knowledge transfer from other capable large language models, the tokenizers (vocabularies) of Gemma (Team et al., 2024), Qwen2 (Yang et al., 2024), LLaMA2 (Touvron et al., 2023b), and LLaMA3 (Meta, 2024) are selected as the target tokenizer to replace. We report hyper-parameters in Appendix A.

**Corpus** To reduce the risk of distribution shift from the training data, we choose the vanilla pre-training corpus (Gao et al., 2020; Soboleva et al., 2023; Kocetkov et al., 2023) of the base model

Pythia in the fine-tuning process. Corpora from downstream tasks and multiple languages are applied in token-level distillation and cross-lingual transfer experiments (Section 4).

**Evaluation Tasks** Following the common practices to evaluate large language models (Lin et al., 2022; Biderman et al., 2023; Zhang et al., 2024), there are 10 datasets, including commonsense reasoning (Conneau et al., 2018; Clark et al., 2018; Mihaylov et al., 2018; Zellers et al., 2019; Ponti et al., 2020; Bisk et al., 2020; Sakaguchi et al., 2020; Tikhonov & Ryabinin, 2021) and reading comprehension (Clark et al., 2019) tasks, used in this work. To avoid the randomness from the prompt and evaluation method, we adopt the default prompt from the commonly used language model evaluation harness framework (Gao et al., 2024).

**Baselines** We introduce the following methods from the cross-lingual vocabulary adaptation domain as baseline methods in this work:

- **Random Initialization** for each token $t \in \{\mathcal{V}_t \setminus (\mathcal{V}_t \cap \mathcal{V}_s)\}$ employs the default initialization method of huggingface transformers and reuses the parameters of token $t \in \{\mathcal{V}_t \cap \mathcal{V}_s\}$, which belongs to overlapping vocabularies.

- **Random Permutation** initializes each token $t \in \{\mathcal{V}_t \setminus (\mathcal{V}_t \cap \mathcal{V}_s)\}$ using the parameter of a randomly chosen token from the source vocabulary. The parameters of shared tokens are also reused.

- **WECHSEL** (Minixhofer et al., 2022) transfers embeddings of source tokens into target tokens by tokenizing and recomposing additional word embeddings $W_s$ and $W_t$, which are aligned with a bilingual dictionary.

- **OFA** (Liu et al., 2024) factorizes the embeddings of source model $E_s$ into the primitive embeddings $P$ and source coordinates $F_s$ that is further re-composed by multilingual word embeddings $W$ to the target coordinates $F_t$. The assembled primitive embeddings $P$ and target coordinates $F_t$ comes the target embeddings $E_t$.

- **Focus** (Dobler & de Melo, 2023) initializes the embedding parameters of token $t \in \{\mathcal{V}_t \setminus (\mathcal{V}_t \cap \mathcal{V}_s)\}$ using the weighted sum of the ones from the token $t \in \{\mathcal{V}_t \cap \mathcal{V}_s\}$. It largely depends on the size of $\| \mathcal{V}_t \cap \mathcal{V}_s \|$, and performs poorly when the overlapping percentage of $\mathcal{V}_t$ and $\mathcal{V}_s$ is low.

- **ZeTT** (Minixhofer et al., 2024) trains an additional hypernetwork $H_\theta$ to generate the parameters for each token $t \in \mathcal{V}_t$. The added hypernetwork brings a lot of training cost.

Table 1: The main results of replacing the vocabulary of Pythia to Gemma using 10B tokens from the Pile corpus. "w/ SlimPajama" adopts 1B tokens from SlimPajama to train GloVe embeddings."+ Align Rep." adds alignment process for GloVe embedding before calculating cosine similarity following Moschella et al. (2023). The best performance among the vocabulary adaptation methods is displayed in **bold**.

| Model | ARC-E 0 | ARC-E 5 | BoolQ 0 | BoolQ 5 | HellaSwag 0 | HellaSwag 5 | OpenbookQA 0 | OpenbookQA 5 | PIQA 0 | PIQA 5 | WinoGrande 0 | WinoGrande 5 | Avg 0 | Avg 5 |
|---|---|---|---|---|---|---|---|---|---|---|---|---|---|---|
| Pythia$_{1B}$ | 56.82 | 58.71 | 60.43 | 57.37 | 37.68 | 37.66 | 18.80 | 19.00 | 70.40 | 71.49 | 53.20 | 52.01 | 49.55 | 49.37 |
| w/ Rand. Init. | 31.36 | 31.61 | 37.83 | 49.11 | 26.35 | 26.40 | 14.00 | 12.60 | 54.57 | 55.33 | 49.17 | 49.17 | 35.55 | 37.37 |
| w/ Rand. Perm. | 31.69 | 32.95 | 37.77 | 54.80 | 26.43 | 26.39 | 14.00 | 12.60 | 55.50 | 55.98 | 47.04 | 50.67 | 35.40 | 38.90 |
| w/ OFA | 38.17 | 37.79 | 55.14 | 52.35 | 28.29 | 28.62 | 14.40 | 12.20 | 58.43 | 58.54 | 49.96 | 50.99 | 40.73 | 40.08 |
| w/ WECHSEL | 43.35 | 45.33 | 56.61 | 54.34 | 32.53 | 32.41 | 14.80 | 16.20 | 61.70 | 62.89 | 52.01 | 52.72 | 43.50 | 43.98 |
| w/ Focus | 46.55 | 48.95 | 56.21 | **55.78** | 32.27 | 32.46 | 19.20 | 18.00 | 63.82 | 64.80 | 51.70 | 51.78 | 44.96 | 45.29 |
| w/ ZeTT | 47.14 | 49.03 | 57.06 | 53.70 | 34.06 | 34.06 | 18.40 | 19.40 | 64.15 | 65.34 | 52.09 | 51.22 | 45.48 | 45.46 |
| w/ UnifyVocab | **54.46** | **56.86** | 58.90 | 52.26 | 36.16 | 36.27 | **21.00** | 20.20 | **67.74** | **68.50** | 52.25 | 50.91 | 48.42 | 47.50 |
| w/ SlimPajama | 53.54 | 55.68 | 57.55 | 53.85 | 36.10 | 35.99 | 19.40 | **20.20** | 67.03 | 67.52 | 52.09 | 51.22 | 47.62 | 47.41 |
| + Align Rep. | 54.25 | 56.65 | **59.33** | 54.68 | **37.08** | **36.91** | 20.20 | 19.40 | 67.36 | 68.17 | **54.38** | **52.80** | **48.77** | **48.10** |
| Pythia$_{2.8B}$ | 63.80 | 67.00 | 63.91 | 65.14 | 45.32 | 45.04 | 24.00 | 25.20 | 74.05 | 74.43 | 58.64 | 60.77 | 54.95 | 56.26 |
| w/ Rand. Init. | 30.47 | 32.91 | 38.20 | 51.07 | 26.46 | 26.69 | 14.40 | 13.20 | 55.17 | 55.06 | 48.30 | 50.51 | 35.50 | 38.24 |
| w/ Rand. Perm. | 31.48 | 31.86 | 37.83 | 50.46 | 26.48 | 26.49 | 13.60 | 14.40 | 54.03 | 54.95 | 50.20 | 48.86 | 35.60 | 37.84 |
| w/ Focus | 54.29 | 58.16 | 61.44 | 62.84 | 38.38 | 39.09 | 20.00 | 20.20 | 68.44 | 68.28 | 54.62 | 56.04 | 49.53 | 50.77 |
| w/ UnifyVocab | **61.62** | **65.15** | **63.82** | **65.47** | **43.13** | **43.18** | **23.40** | **25.80** | **72.14** | **72.42** | **58.17** | **61.17** | **53.71** | **55.53** |

## 3.2 MAIN RESULTS AND ANALYSES

We first conduct experiments to replace the tokenizer of Pythia with the Gemma tokenizer using 10B tokens. Results on six datasets are shown in Table 1. Given limited tokens to fine-tune, it can be found that UnifyVocab performs better than the other three baseline methods. The average improvement of UnifyVocab over the strong baseline method ZeTT reaches 2.49%, and the 97.63% performance of the vanilla model is reserved after vocabulary replacement. Replace the corpus to train the GloVe embedding with 1B SlimPajama (Soboleva et al., 2023) tokens comes to a comparable results. It demonstrates the robustness of our method on the pre-training corpus for token embedding and alignment matrix. We find that the performance can be further advanced by aligning the GloVe embedding into the relative representation using 300 common tokens occur in both vocabularies following Moschella et al. (2023), which is the row with "+ Align Rep." label.

**Better alignment brings better initialization.** The loss curves of Pythia$_{1B}$ with different methods during the first 5B tokens training are shown in Figure 3(a). We find that UnifyVocab brings a better initialization and decreases the first-step training loss from 17.8 (Focus) to 9.5. Moreover, the training process with UnifyVocab is faster than the ones with other methods, which reaches 2.75 with only 2.6B tokens and is 1.92x (5B/2.6B) speed up than the method Focus.

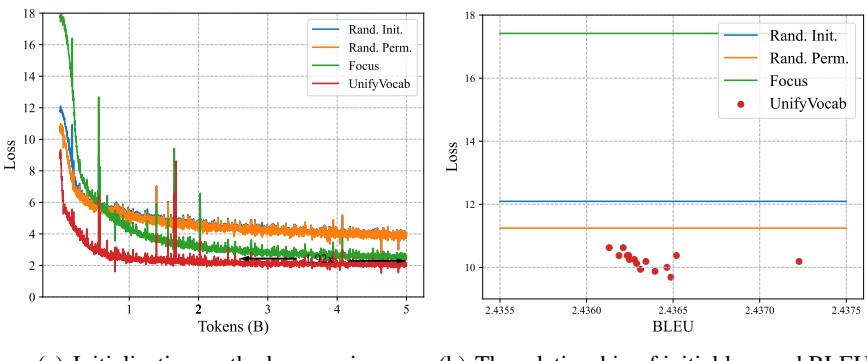

(a) Initialization method comparison     (b) The relationship of initial loss and BLEU

Figure 3: The training loss of Pythia$_{2.8b}$ with different methods (a) and $M_{s \to t}$ learned using UnifyVocab, which is denoted by red point (b).

Table 2: The benchmark results of UnifyVocab using 10B tokens from the Pile corpus. The overlapping ratio between the vocabulary of Pythia and other models are 6.23%(Gemma), 26.92%(Qwen2), 28.10%(LLaMA2), 32.85%(LLaMA3).

| Model | #$\mathcal{V}$ (k) | ARC-E 0 | ARC-E 5 | BoolQ 0 | BoolQ 5 | HellaSwag 0 | HellaSwag 5 | OpenbookQA 0 | OpenbookQA 5 | PIQA 0 | PIQA 5 | WinoGrande 0 | WinoGrande 5 | Avg 0 | Avg 5 |
|---|---|---|---|---|---|---|---|---|---|---|---|---|---|---|---|
| Pythia$_{1B}$ | 50.3 | 56.82 | 58.71 | 60.43 | 57.37 | 37.68 | 37.66 | 18.80 | 19.00 | 70.40 | 71.49 | 53.20 | 52.01 | 49.55 | 49.37 |
| → Gemma | 256.0 | 54.46 | 56.86 | **58.90** | 52.26 | 36.16 | 36.27 | **21.00** | 20.20 | 67.74 | 68.50 | 52.25 | 50.91 | 48.42 | 47.50 |
| → Qwen2 | 152.1 | 54.46 | 57.07 | 54.80 | 49.79 | 37.18 | 37.04 | 19.20 | 18.40 | 68.44 | **70.24** | 53.35 | 52.80 | 47.91 | 47.56 |
| → LLaMA2 | 32.0 | 49.45 | 52.02 | 58.32 | 55.75 | 35.38 | 35.45 | 18.80 | 17.80 | 66.32 | 66.65 | 53.91 | 50.91 | 47.03 | 46.43 |
| → LLaMA3 | 128.0 | **54.63** | **57.28** | 55.84 | **53.70** | **37.34** | **37.43** | 20.20 | **20.40** | 69.04 | 70.18 | **54.46** | **53.43** | **48.59** | **48.74** |
| Pythia$_{2.8B}$ | 50.3 | 63.80 | 67.00 | 63.91 | 65.14 | 45.32 | 45.04 | 24.00 | 25.20 | 74.05 | 74.43 | 58.64 | 60.77 | 54.95 | 56.26 |
| → Gemma | 256.0 | 61.62 | 65.15 | 63.82 | **65.47** | 43.13 | 43.18 | 23.40 | **25.80** | 72.14 | 72.42 | 58.17 | **61.17** | 53.71 | **55.53** |
| → Qwen2 | 152.1 | **62.54** | **66.04** | 62.35 | 63.55 | 44.46 | 44.39 | 23.20 | 24.60 | **73.50** | **73.56** | **59.04** | 59.59 | 54.18 | 55.29 |
| → LLaMA3 | 128.0 | 61.83 | 64.60 | **64.40** | 63.94 | **44.62** | **44.59** | 23.80 | 25.60 | 73.45 | 73.29 | 57.54 | 58.72 | **54.27** | 55.12 |
| Pythia$_{6.9B}$ | 50.3 | 65.99 | 69.23 | 62.84 | 62.02 | 47.56 | 47.64 | 25.00 | 27.00 | 74.65 | 75.41 | 60.46 | 62.43 | 56.08 | 57.29 |
| → Gemma | 256.0 | 65.40 | 68.35 | 62.39 | 59.57 | 45.75 | 45.86 | 22.00 | 25.60 | 73.39 | 74.10 | 60.38 | 61.17 | 54.89 | 55.77 |
| → Qwen2 | 152.1 | 65.57 | **68.43** | **64.07** | 57.61 | 46.84 | 46.91 | **25.60** | 25.40 | 73.45 | 74.65 | 61.17 | 63.14 | 56.12 | 56.02 |
| → LLaMA3 | 128.0 | **66.46** | 68.35 | 63.79 | 60.64 | **47.28** | 47.31 | 25.60 | **28.20** | **74.48** | **75.84** | 61.48 | 63.30 | **56.52** | **57.27** |

We further investigate the impact of the learned alignment matrix $M_{s \to t}$ by changing the hyperparameters of GloVe. It is noted that different alignment matrices $M_{s \to t}$ bring different initial parameters, and also come to different BLEU scores on the same evaluation corpus. Figure 3(b) illustrates the negative relationship between the first-step training loss and the BLEU. In other words, the higher the BLEU score for the alignment matrix $M_{s \to t}$, the better the initial parameter is. The other metrics like BLEU-1 and BertScore are also used to evaluate the alignment metrix, and also show a negative relationship with the initial training loss in Appendix B.4.

**More overlapping comes to faster convergence and higher performance.**  The UnifyVocab is further applied to the other three target tokenizers: Qwen2, LLaMA2, and LLaMA3. Table 2 reports the performance of models after replacing vocabulary on six datasets. UnifyVocab recovers 98.02% performance of the base model on average with only 10B tokens. Given a target vocabulary with more tokens than the one of Pythia (50.3k), it can be found that a higher overlapping ratio brings a better performance of model replaced (97.62% for Gemma to 99.07% for LLaMA3). The zero-shot in-context learning results for $\text{Pythia}_{6.9B}$ with LLaMA3 vocabulary even surpass the vanilla base model. The results of $\text{Pythia}_{1B}$ with LLaMA2 vocabulary are only 94.47%, which is inferior to the average result of 98.02%. We argue that it may come from the missing 75.0M parameters (7.4% for $\text{Pythia}_{1B}$) after switching to a 32.0k vocabulary from the 50.3k vocabulary.

Figure 4(a) shows the training loss curve during the first 5B tokens. The replacing process of the Gemma tokenizer is the slowest, which may come from the only 6.23% overlapping ratio between two vocabularies. It is interesting to find that other conditions for three tokenizers converge with only 1B tokens under the same setting. Further analyses for the convergence of vocabulary adaptation refer to Appendix B.2, which shows a similar phenomenon.

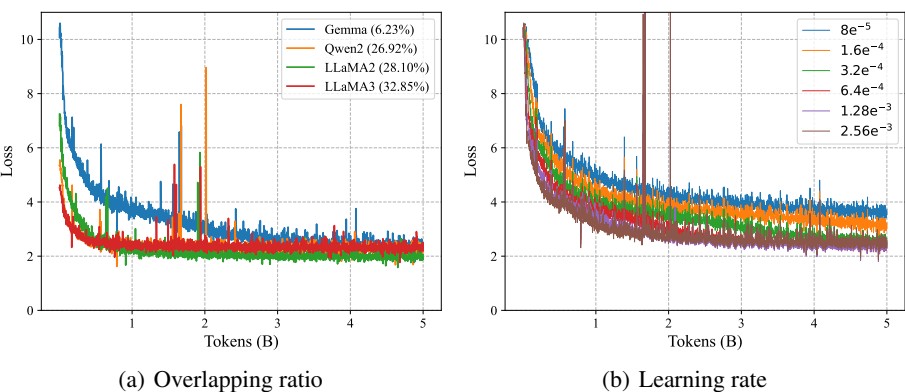

(a) Overlapping ratio                    (b) Learning rate

Figure 4: The training loss curve of $\text{Pythia}_{1B}$ for different overlapping ratios (a), and learning rate used during replacing to the Gemma tokenizer (b).

**Two-stage tuning brings a more stable convergence.**  To replace the tokenizer and keep the performance of the vanilla model, we adopt only fine-tuning the vocabulary-related parameters at the first stage. The main reason for two-stage tuning is to take these parameters as the adapters for different tokenizers, and avoid the well-trained parameters of the internal layer distracted by the new initialized parameters.

Figure 5 illustrates that our two-stage tuning method makes the convergence more stable under a high learning rate like $6.4e^{-4}$, which comes to better performance after tuning on 10B tokens. It is noted that the loss spike also occurs at the first stage, fine-tuning vocabulary-related parameters only, under such a high learning rate like $2.56e^{-3}$ in Figure 4(b).

## 4    APPLICATIONS

In this section, we illustrate two direct applications of UnifyVocab: token-level distillation (Section 4.1) and cross-lingual knowledge transfer (Section 4.2).

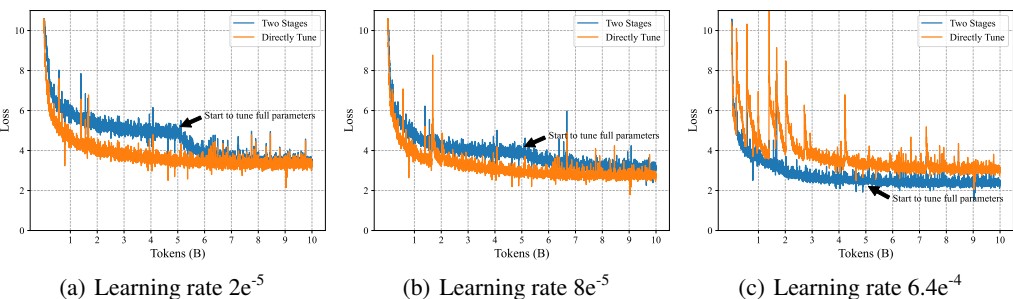

| (a) Learning rate 2e$^{-5}$ | (b) Learning rate 8e$^{-5}$ | (c) Learning rate 6.4e$^{-4}$ |

Figure 5: The loss curve of Pythia$_{1B}$ under two-stage tuning or direct full parameters tuning.

### 4.1 TOKEN-LEVEL DISTILLATION

To compensate for the performance gap between these capable open-source language models and Pythia, we take these models as the teacher model of Pythia after replacing the tokenizer. Training samples from downstream tasks and the corpus of Pile are used in the token-level distillation experiments. The logit of each token from the teacher model is taken as the soft label of Pythia to learn. We empirically set the proportion of training samples to 15% to avoid a significant degradation in the performance of language modeling (Wei et al., 2023).

Table 3 reports the results of two baseline methods and token-level distillation from three teacher models using 235M tokens. We can find that token-level distillation is significantly better than the one of sentence-level distillation. Given the same teacher model Qwen2$_{7B}$, the improvement of Pythia over the sentence-level distillation result reaches 4.37%, which further demonstrates the importance of unifying tokenizer between models. The knowledge transfer between models will be constrained in sentence-level distilling without unifying vocabulary. It is also noted that models with token-level distillation on strong teacher models like Qwen2 outperform the ones of direct tuning.

Table 3: The main results of token-level distillation on six downstream tasks using 235M tokens. "+Sentence distill" denotes the sentence-level distillation results with Qwen2$_{7B}$(Yang et al., 2024), which fine-tunes on the output from Qwen2$_{7B}$ given questions as prompt.

| | ARC-E | | BoolQ | | HellaSwag | | OpenbookQA | | PIQA | | WinoGrande | | Avg | |
| --- | --- | --- | --- | --- | --- | --- | --- | --- | --- | --- | --- | --- | --- | --- |
| **Model** | 0 | 5 | 0 | 5 | 0 | 5 | 0 | 5 | 0 | 5 | 0 | 5 | 0 | 5 |
| Pythia$_{1B}$ | 56.82 | 58.71 | 60.43 | 57.37 | 37.68 | 37.66 | 18.80 | 19.00 | 70.40 | 71.49 | 53.20 | 52.01 | 49.55 | 49.37 |
| + Direct tuning | 57.49 | 55.64 | 70.70 | 72.11 | 41.24 | 41.60 | 25.40 | 28.40 | 69.04 | 70.08 | 54.70 | 54.78 | 53.10 | 53.77 |
| + Sentence distill | 52.27 | 53.41 | 67.49 | 67.06 | 39.03 | 39.08 | 21.80 | 22.80 | 66.97 | 68.99 | 51.85 | 52.17 | 49.90 | 50.58 |
| w/ Gemma$_{7B}$ | 55.39 | 56.99 | 67.19 | 69.69 | 36.53 | 37.26 | 19.00 | 22.80 | 68.82 | 69.21 | 52.33 | 53.51 | 49.88 | 51.58 |
| w/ Qwen2$_{7B}$ | 62.33 | 63.17 | 70.18 | 72.54 | 41.58 | 42.21 | 22.00 | **28.20** | **73.01** | 73.18 | 55.01 | 55.56 | 54.02 | 55.81 |
| w/ LLaMA3$_{8B}$ | **64.02** | **64.56** | **73.91** | **74.19** | **42.11** | **42.34** | **24.20** | 27.60 | 72.74 | **73.83** | **55.49** | **56.43** | **55.41** | **56.49** |
| Pythia$_{6.9B}$ | 65.99 | 69.23 | 62.84 | 62.02 | 47.56 | 47.64 | 25.00 | 27.00 | 74.65 | 75.41 | 60.46 | 62.43 | 56.08 | 57.29 |
| + Direct tuning | 66.25 | 66.20 | 79.30 | 78.87 | 52.21 | 53.39 | 33.20 | 33.00 | 72.91 | 74.48 | 62.90 | 61.72 | 61.13 | 61.28 |
| + Sentence distill | 61.70 | 65.36 | 76.64 | 76.88 | 48.98 | 51.33 | 28.20 | 30.40 | 70.18 | 71.55 | 58.96 | 62.19 | 57.44 | 59.62 |
| w/ Gemma$_{7B}$ | 67.59 | 68.94 | 76.06 | 75.66 | 47.83 | 48.36 | 28.40 | 31.40 | 73.78 | 75.52 | 59.04 | 64.17 | 58.78 | 60.67 |
| w/ Qwen2$_{7B}$ | **71.72** | **73.27** | **79.85** | **80.00** | **50.78** | **51.12** | **29.20** | **34.00** | **77.26** | **77.91** | **61.33** | **64.56** | **61.69** | **63.48** |
| w/ LLaMA3$_{8B}$ | 67.05 | 69.78 | 77.83 | 78.78 | 48.83 | 50.15 | 26.00 | 32.00 | 74.21 | 76.22 | 60.22 | 60.93 | 59.02 | 61.31 |

### 4.2 CROSS-LINGUAL TRANSFER

The tokens for other languages can be aligned and initialized by the tokens with similar semantics in the source vocabulary, which can speed up the cross-lingual knowledge transfer. In this section, we use UnifyVocab to conduct cross-lingual transfer experiments using 4B tokens from the CulturaX corpus. The tokenizer of Qwen2 is used as the target tokenizer for Pythia.

Table 4: The normalized perplexity on the valid corpus of CulturaX. The perplexity is normalized to the vocabulary of Pythia following Wei et al. (2023). "**High**", "**Medium**" and "**Low**" denotes the available amount of linguistic resources.

| Model | #Tune (B) | High | | | | | Medium | | | | | Low | | | Avg |
|---|---|---|---|---|---|---|---|---|---|---|---|---|---|---|---|
| | | ar | de | en | ja | zh | bn | ko | th | uk | vi | ta | te | ur | |
| Qwen2$_{1.5B}$ | – | 4.7 | 11.1 | 15.7 | 6.0 | 4.6 | 2.4 | 3.3 | 2.6 | 5.7 | 3.3 | 2.8 | 3.4 | 4.0 | 5.3 |
| Pythia$_{1B}$ | – | 7.6 | 15.4 | **21.7** | 9.9 | 13.2 | 3.4 | 5.6 | 4.3 | 6.7 | 6.3 | 2.9 | 3.3 | 5.8 | 8.2 |
| w/ Focus | 0 | $4.1e^3$ | $1.7e^5$ | $1.8e^6$ | $2.1e^4$ | $9.6e^2$ | $6.5e^4$ | $1.0e^3$ | $5.6e^3$ | $1.6e^6$ | $8.4e^2$ | $5.0e^4$ | $1.9e^5$ | $1.9e^5$ | $3.1e^5$ |
| | 4 | 8.3 | 27.1 | 59.7 | 14.0 | 14.0 | 3.6 | 5.9 | 3.8 | 7.3 | 5.9 | 3.5 | 3.6 | 4.3 | 12.4 |
| w/ UnifyVocab | 0 | $1.6e^2$ | $9.4e^2$ | $3.6e^2$ | $3.1e^2$ | $1.5e^2$ | 89.6 | 94.1 | 94.3 | $1.6e^2$ | $1.1e^2$ | 36.1 | 27.8 | $1.1e^2$ | $2.0e^2$ |
| | 4 | **6.5** | **14.1** | 24.0 | **9.0** | **9.2** | **2.5** | **4.5** | **3.2** | **5.3** | **4.5** | **2.3** | **2.4** | **3.8** | **7.0** |
| Qwen2$_{7B}$ | – | 3.9 | 8.1 | 11.8 | 4.9 | 3.8 | 2.1 | 2.9 | 2.3 | 3.8 | 2.9 | 2.3 | 2.6 | 3.3 | 4.2 |
| Pythia$_{6.9B}$ | – | 5.9 | 10.8 | **16.7** | 7.9 | 9.9 | 3.0 | 4.6 | 3.7 | 4.9 | 4.9 | 2.6 | 2.9 | 4.8 | 6.3 |
| w/ Focus | 0 | $6.9e^3$ | $1.6e^5$ | $1.2e^6$ | $2.4e^4$ | $1.3e^3$ | $2.5e^4$ | $7.2e^2$ | $3.3e^3$ | $1.9e^6$ | $7.9e^2$ | $1.7e^4$ | $1.5e^5$ | $1.2e^5$ | $2.8e^5$ |
| | 4 | 6.8 | 17.6 | 39.3 | 10.8 | 11.1 | 2.5 | 5.0 | 3.3 | 5.2 | 4.8 | 2.3 | 2.5 | 3.7 | 8.8 |
| w/ UnifyVocab | 0 | $1.9e^2$ | $8.0e^2$ | $2.8e^2$ | $3.3e^2$ | $1.6e^2$ | 85.3 | 97.0 | 94.3 | $1.7e^2$ | $1.1e^2$ | 36.1 | 23.8 | $1.0e^2$ | $1.9e^2$ |
| | 4 | **5.4** | **10.1** | 18.1 | **7.5** | **8.0** | **2.1** | **4.0** | **2.8** | **4.1** | **3.8** | **2.1** | **2.1** | **3.1** | **5.6** |

As shown in Table 4, the perplexity of Pythia initialized with UnifyVocab ($2.0e^2$) is significantly better than the one of Focus baseline ($2.9e^5$). After only 4B tokens tuning, the improvement of UnifyVocab is 13.1% over the vanilla model on average. The performance of Pythia using UnifyVocab on three low-resource languages even outperforms the ones of Qwen2 under a similar parameter amount.

Table 5 and 7 report in-context learning results on four multilingual datasets. We can find that UnifyVocab brings a better-initialized model than the baseline method Focus (+3.5%), and transfers the knowledge into other languages like Vietnamese (vi, +2.3%) and Urdu (ur, +0.9%).

It is interesting to find that the perplexity of Pythia$_{1B}$ initialized by UnifyVocab reaches $2.0e^2$, while the in-context learning results are comparable with the ones of Focus after 4B tokens tuning. We argue that it arises from the mostly reserved English ability with UnifyVocab, which is 56.2% outperforming the 43.6% of Focus.

Table 5: Zero-shot in-context learning results of cross-lingual transfer. "#Tune(B)=0" denotes performance of the model after parameter initialization without any tuning. Refer to Table 7 in Appendix B.5 for five-shot results.

| Model | #Tune(B) | XNLI | | | | | | | XCOPA | | | | XStoryCloze | | | | XWinograd | | | Avg |
|---|---|---|---|---|---|---|---|---|---|---|---|---|---|---|---|---|---|---|---|---|
| | | en | de | zh | ar | th | vi | ur | en | th | vi | ta | en | zh | ar | te | en | zh | ja | |
| Pythia$_{1B}$ | – | 51.0 | 37.8 | **42.6** | 35.9 | 34.8 | 37.0 | 34.7 | **62.4** | 54.4 | 50.6 | 55.4 | **64.4** | 48.7 | 48.0 | 52.9 | 57.1 | **53.2** | **59.3** | **48.9** |
| w/ Focus | 0 | 32.8 | 32.2 | 33.6 | 33.6 | 33.5 | 32.0 | 32.8 | 49.4 | 51.2 | 48.4 | 54.4 | 46.0 | 47.7 | **48.7** | 46.5 | 49.7 | 47.2 | 50.3 | 42.8 |
| | 4 | 46.0 | 35.1 | 34.9 | 32.9 | 32.5 | 35.4 | 34.7 | 53.0 | 52.6 | 50.0 | 54.2 | 57.1 | 50.0 | 47.5 | 52.5 | 52.2 | 51.7 | 54.4 | 45.9 |
| w/ UnifyVocab | 0 | 48.4 | 35.9 | 33.4 | 33.1 | 31.8 | 32.5 | 33.8 | 54.6 | 52.0 | 47.4 | **57.2** | 58.6 | 46.5 | 46.7 | 51.0 | 54.4 | 50.2 | 50.5 | 45.4 |
| | 4 | **51.2** | **39.0** | 42.3 | **38.5** | **35.8** | **38.9** | **35.7** | 60.8 | **55.2** | **51.8** | 53.8 | 64.0 | **51.0** | 47.5 | **54.1** | 56.0 | 52.5 | 56.9 | **49.2** |
| Pythia$_{6.9B}$ | – | 54.4 | 39.0 | 46.2 | 39.3 | 39.8 | 39.3 | 36.4 | 70.8 | 57.6 | 51.2 | 53.0 | 70.7 | 54.0 | 50.4 | 53.5 | 63.7 | 60.1 | 67.1 | 52.6 |
| w/ Focus | 0 | 31.5 | 31.3 | 33.0 | 32.6 | 33.4 | 32.2 | 32.6 | 46.4 | 52.4 | 49.0 | **56.6** | 44.6 | 47.3 | 48.2 | 47.4 | 48.3 | 46.8 | 51.1 | 42.5 |
| | 4 | 52.6 | 34.9 | 36.6 | 35.1 | 33.6 | 39.0 | 34.5 | 61.6 | 52.4 | 52.0 | 53.8 | 62.1 | 49.3 | 47.1 | **54.6** | 56.2 | 52.1 | 58.9 | 48.1 |
| w/ UnifyVocab | 0 | 50.9 | **37.6** | 34.3 | 34.6 | 33.7 | 33.1 | 33.7 | 60.2 | 52.6 | 48.0 | 55.8 | 63.1 | 47.1 | 47.0 | 50.3 | 59.6 | 48.6 | 51.4 | 46.8 |
| | 4 | **55.1** | 35.5 | **41.6** | **39.1** | **39.6** | **42.8** | **37.1** | **70.2** | **56.0** | **53.6** | 51.4 | **70.4** | **52.5** | **49.1** | 54.3 | **61.5** | 54.0 | **60.7** | **51.4** |

## 5 RELATED WORKS

Our work is related to word representation, large language models, and vocabulary adaption, which will be briefly introduced below.

**Word Representation** Based on the distributional semantic hypothesis, Bengio et al. (2003) introduced the neural probabilistic language model to learn word representation. Researchers mainly

focus on improving the effectiveness during learning word representations (Mikolov et al., 2013a;b; Bojanowski et al., 2017), which provide a good initialization for neural networks like LSTM and GRU (Hochreiter, 1997; Chung et al., 2014). GloVe (Pennington et al., 2014) provides a method to train word representations from a view of global word-word co-occurrence matrix decomposition. It motivates us to train a word representation for each token and align the token ID from statistical co-occurrence information in the pre-training corpus.

**Large Language Model**   Through scaling in the parameters and pre-training corpus (Kaplan et al., 2020; Hoffmann et al., 2022), large language models including GPT and LLaMA (Radford et al., 2018; 2019; Brown et al., 2020; OpenAI, 2023; Touvron et al., 2023a;b; Meta, 2024; GLM et al., 2024) demonstrate impressive performance across multiple tasks. However, the knowledge transfer between different models is greatly hindered by the mismatch in the vocabulary. We aim to mitigate this problem by introducing an effective method to replace the tokenizer of a pre-trained large language model.

**Vocabulary Adaption**   is investigated mainly in the multilingual domain, especially the cross-lingual knowledge transfer problem (Workshop et al., 2023; Muennighoff et al., 2023; Yang et al., 2023; Zhu et al., 2023; Üstün et al., 2024; Li et al., 2024). It aims to improve the encoding effectiveness of tokenizer on corpora from new languages, and is often implemented by extending the original vocabulary (Tran, 2020; Chau et al., 2020; Minixhofer et al., 2022; Dobler & de Melo, 2023; Downey et al., 2023). Most methods like Focus (Dobler & de Melo, 2023) rely on the tokens belonging to both source vocabulary and target vocabulary to initialize the other new tokens in the target vocabulary. Our method differs from these studies for the whole replacement of vocabulary using a limited corpus. It does not rely on the tokens in both source vocabulary and target vocabulary.

## 6    LIMITATIONS

The first limitation comes from the assumption that the pre-training data distribution is available. We conduct experiments on Pythia with different parameter amounts, which provide public model weights and pre-training corpus. Due to the limited computation resource budget, open-source language models with unknown pre-training corpus like Mistral (Jiang et al., 2023) are not investigated in this work. However, the pre-training corpus distribution of open-weighted large language models can be roughly inferred by the BPE vocabulary (Hayase et al., 2024). It can re-construct a similar pre-training corpus to conduct replacing tokenizer experiments.

The 10B tokens of model tuning cost in replacing a tokenizer using UnifyVocab is another limitation, although it is only 3.33% of the 300B tokens pre-training corpus for Pythia. From the loss curve of UnifyVocab (Figure 4), we find that the start of full parameters tuning can be less than 5B tokens, which may result in a better balance.

## 7    CONCLUSION AND FUTURE WORK

In this paper, we introduce a method named UnifyVocab to replace the tokenizer of large language models from a token-token co-occurrence view. Extensive experiments demonstrate that UnifyVocab reserves the most performance of vanilla models (98.02% on average) using only 10B tokens, which enables deeper knowledge transfer between models like token-level distillation and cross-lingual knowledge transfer.

Beyond replacing the vocabulary of large language models, our method can be extended to replace the vocabulary of multi-modal models by aligning different modal tokens. The other direction is to develop a method with less training cost, e.g., incorporating meta-learning to replace the two-stage tuning method.

## 8    REPRODUCIBILITY STATEMENT

Codes and weights will be made public after review to advocate future research. Hyper-parameters are reported in the Appendix A. The weight of models with replaced vocabulary and source codes will be public after review to advocate future research.

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

## A  HYPER-PARAMETERS

**GloVe Training**   We empirically train GloVe vectors with 1B tokens, which covers most tokens from Gemma (95.10%), Qwen2 (93.40%), LLaMA2 (99.35%), and LLaMA3 (98.04%). The dimension size is set to 300. The max training iteration and the size of the slide window are 15.

**Model Tuning**   The optimizer adopted in this work is AdamW (Loshchilov & Hutter, 2019), where $\beta_1 = 0.9$ and $\beta_2 = 0.999$. The learning rate for baseline methods is set to 5e-5 to reduce the loss spike in Figure 5(b) and Figure 5(c). We adopt bf16 mixed precision training and ZeRO-1 to save GPU memory cost and speed up the training process (Micikevicius et al., 2018; Rasley et al., 2020). Following Biderman et al. (2023), the batch size is set to 2M tokens and the max sequence length is 2048.

## B  ADDITIONAL RESULTS

### B.1  GLOVE VECTORS

We show the effects of different token amounts for the GloVe vectors training in Figure 6. It can be found that 1B tokens used in this work provide a high vocabulary coverage (>90%) and better initialization for Pythia$_{1B}$. Due to the limited computation budget, experiments with more than 1B tokens are not conducted.

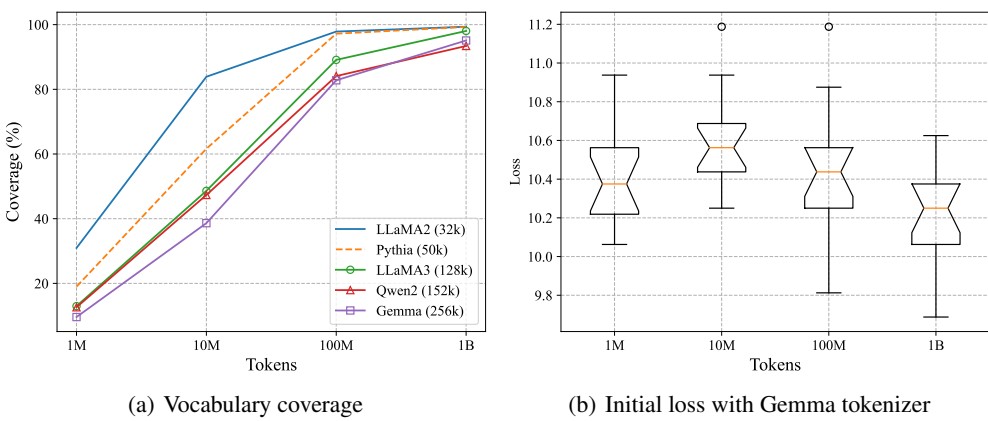

(a) Vocabulary coverage  (b) Initial loss with Gemma tokenizer

Figure 6: The average vocabulary coverage (a) and initial training loss of Pythia$_{1B}$ (b) under different amount tokens to train the GloVe vector.

## B.2 CONVERGENCE ANALYSIS

To investigate the effect of overlapping rate between two tokenizers to the convergence of training, we plot Figure 7(a) for the random initialization baseline method. The convergence of Gemma tokenizer is slower than the other tokenizers and comes to worse results, which are similar to the case in 4(a). Moreover, we randomly shuffle the alignment matrix learned in UnifyVocab to imitate the case that other worse methods rather than cosine similarity to calculate the alignment matrix. Figure 7(b) shows that the higher percentage of randomly shuffle comes to higher initial training loss and slower convergence.

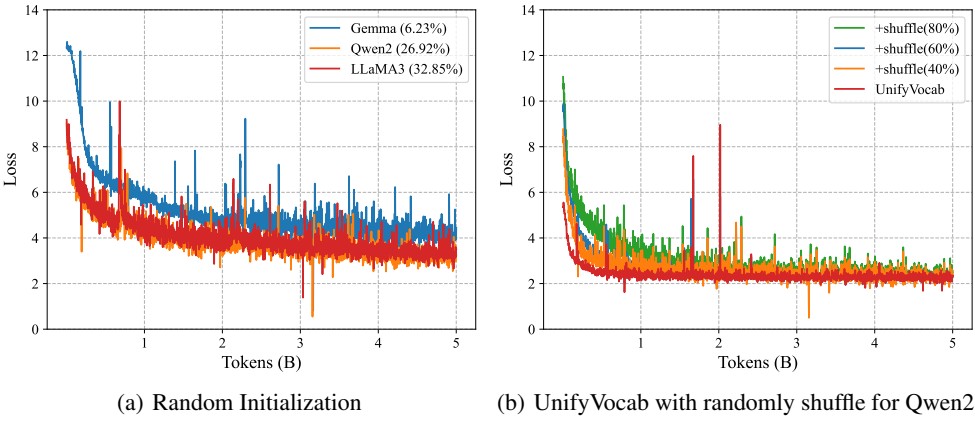

(a) Random Initialization  (b) UnifyVocab with randomly shuffle for Qwen2

Figure 7: The training loss for random initialization to different tokenizers (a) and UnifyVocab for Qwen2 using Pythia$_{1b}$.

## B.3 VOCABULARY ADAPTATION RESULTS WITH 2B TOKENS

We further investigate a challenge condition that only 2B tokens are provided to adapt the target vocabulary. To meet the requirement, batch size is set to 1M tokens and training steps are reduced to 2k, correspondingly. Table 6 shows results of adapting to other 3 tokenizers using UnifyVocab. It can be found that 95.66% performance of vanilla model is recovered on average, which further demonstrates the effectiveness of our method.

## B.4 ADDITIONAL ALIGNMENT METRICS

The BLEU-1 and BertScore can also be used to evaluate the performance of alignment matrix learned. The alignment evaluation process of BLEU-1 is same with the one of BLEU, which is the averaged of

Table 6: The main results of replacing the vocabulary of Pythia for UnifyVocab using 2B tokens from the Pile corpus.

| Model | #$\mathcal{V}$ (k) | ARC-E 0 | 5 | BoolQ 0 | 5 | HellaSwag 0 | 5 | OpenbookQA 0 | 5 | PIQA 0 | 5 | WinoGrande 0 | 5 | Avg 0 | 5 |
|---|---|---|---|---|---|---|---|---|---|---|---|---|---|---|---|
| Pythia$_{1B}$ | 50.3 | 56.82 | 58.71 | 60.43 | 57.37 | 37.68 | 37.66 | 18.80 | 19.00 | 70.40 | 71.49 | 53.20 | 52.01 | 49.55 | 49.37 |
| → Gemma | 256.0 | 51.09 | 52.44 | 53.12 | 52.35 | 35.00 | 35.05 | 20.20 | 18.60 | 64.80 | 65.83 | 53.12 | 51.62 | 46.22 | 45.98 |
| → Qwen2 | 152.1 | 53.41 | 55.47 | 53.52 | 55.81 | 36.12 | 36.38 | 20.80 | 18.00 | 68.50 | 68.88 | 54.38 | 52.80 | 47.79 | 47.89 |
| → LLaMA3 | 128.0 | 51.73 | 55.09 | 59.05 | 55.08 | 36.42 | 36.52 | 19.40 | 19.60 | 67.68 | 68.34 | 53.43 | 53.75 | 47.95 | 48.06 |

BLEU-1, BLEU-2, BLEU-3 and BLEU-4. As for BertScore, we first de-tokenized the target token ID corpus $\mathcal{C}'_t$ using Tokenizer$_t$ into the text corpus $\mathcal{C}'$, and evaluate the semantic similarity between $\mathcal{C}'$ and the vanilla test corpus $\mathcal{C}$ using the sentence embedding model named "all-mpnet-base-v2" (Song et al., 2020). As shown in Figure 8, these metrics both show a clear negative relationship with the inital training loss.

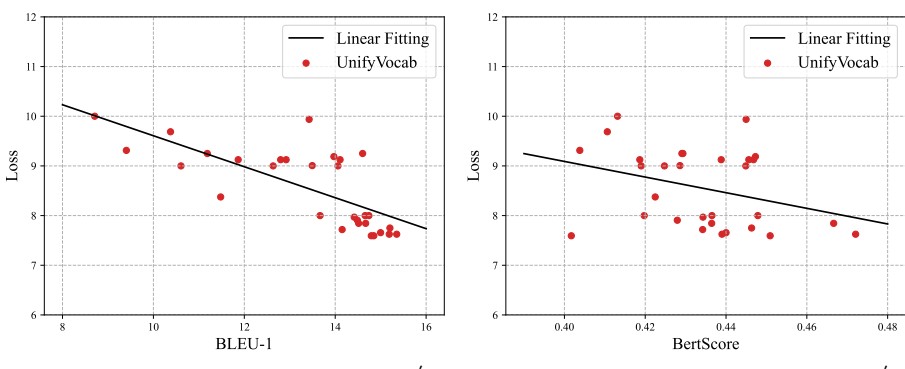

(a) Initial training loss and BLEU-1($\mathcal{C}_t, \mathcal{C}'_t$)    (b) Initial training loss and BertScore($\mathcal{C}, \mathcal{C}'$)

Figure 8: The relationship between initial training loss and BLEU-1 (a) or BertScore (b) for Pythia$_{1b}$.

## B.5 CROSS-LINGUAL TRANSFER

Table 7 reports the 5-shot in-context learning results on 4 multilingual datasets. The average improvement over the baseline method Focus is 3.4% after 4B tokens tuning. We can find that the model initialized by UnifyVocab is comparable to the one of Focus after 4B tokens tuning.

Table 7: The 5-shot in-context learning results of cross-lingual transfer.

| Model | #Tune(B) | XNLI en | de | zh | ar | th | vi | ur | XCOPA en | th | vi | ta | XStoryCloze en | zh | ar | te | XWinograd en | zh | ja | Avg |
|---|---|---|---|---|---|---|---|---|---|---|---|---|---|---|---|---|---|---|---|---|
| Pythia$_{1B}$ | − | 46.2 | 38.6 | 38.9 | 36.9 | 35.2 | 38.9 | 34.9 | 64.0 | 54.0 | 49.4 | 55.2 | 65.5 | 48.4 | 48.2 | 53.0 | 68.9 | 59.7 | 51.4 | 49.3 |
| w/ Focus | 0 | 32.8 | 32.2 | 33.6 | 33.6 | 33.5 | 32.0 | 32.8 | 49.4 | 51.2 | 48.4 | 54.4 | 46.0 | 47.7 | **48.7** | 46.5 | 49.7 | 47.2 | 50.3 | 42.8 |
|  | 4 | 47.0 | 36.7 | 35.4 | 34.3 | 33.5 | 35.1 | 33.9 | 54.2 | 52.2 | 51.6 | 54.8 | 57.0 | 50.4 | 47.6 | 52.2 | 55.4 | 53.8 | 50.9 | 46.4 |
| w/ UnifyVocab | 0 | **48.4** | 35.9 | 33.4 | 33.1 | 31.8 | 32.5 | 33.8 | 54.6 | 52.0 | 47.4 | **57.2** | 58.6 | 46.5 | 46.7 | 51.0 | 54.4 | 50.2 | 50.5 | 45.4 |
|  | 4 | 44.5 | **37.5** | **38.3** | **35.6** | **35.0** | **37.7** | **35.5** | **63.4** | **54.4** | **52.0** | 53.8 | **65.0** | **51.2** | 48.1 | **53.3** | **65.8** | **58.7** | **53.3** | **49.1** |
| Pythia$_{6.9B}$ | − | 53.0 | 40.7 | 41.7 | 38.9 | 37.3 | 41.3 | 35.1 | 75.2 | 58.0 | 54.2 | 52.4 | 73.9 | 54.1 | 50.4 | 54.0 | 73.6 | 71.0 | 56.8 | 53.4 |
| w/ Focus | 0 | 31.5 | 31.3 | 33.0 | 32.6 | 33.4 | 32.2 | 32.6 | 46.4 | 52.4 | 49.0 | **56.6** | 44.6 | 47.3 | 48.2 | 47.4 | 48.3 | 46.8 | 51.1 | 42.5 |
|  | 4 | 45.1 | 37.7 | 35.3 | 33.4 | 35.0 | 38.1 | 33.8 | 58.8 | 53.8 | 51.6 | 53.2 | 63.2 | 50.0 | 46.7 | **54.5** | 61.7 | 62.5 | 52.2 | 48.1 |
| w/ UnifyVocab | 0 | **50.9** | 37.6 | 34.3 | 34.6 | 33.7 | 33.1 | 33.7 | 60.2 | 52.6 | 48.0 | 55.8 | 63.1 | 47.1 | 47.0 | 50.3 | 59.6 | 48.6 | 51.4 | 46.8 |
|  | 4 | 46.8 | **39.1** | **37.3** | **37.7** | **38.0** | **42.5** | 34.9 | **73.2** | **55.6** | **54.6** | 53.4 | **73.1** | **53.9** | **49.2** | 54.0 | **74.0** | **63.3** | **56.7** | **52.1** |

**Case study of multilingual token alignment.** Table 8 provides nine new tokens from three languages with their top 3 tokens in the source vocabulary. In most cases, a clear semantic relationship

between two aligned tokens cannot be found. We argue that it may come from the following two reasons:

Table 8: The case study of new tokens from other languages in the target vocabulary with top-3 source tokens aligned. The language family of French, Chinese, and Korean are Indo-European, Sino-Tibetan, and Koreanic, respectively.

| Top-3 | French | | | Chinese | | | Korean | | |
| | dire(speak) | aller(go) | oui(are) | 吃(eat) | 科学(science) | 智能(intelligence) | 능(competence) | 집(house) | 왜(why) |
|---|---|---|---|---|---|---|---|---|---|
| | | | | *Qwen2 (Target Tokenizer)* | | | | | |
| 1 | ada | Ġsta | Ġsalv | allel | Ġantagon | _{[ | Si | ĠBart | bst |
| 2 | ays | ĠÃ¨ | Ġvas | Ġindicator | Ġign | liquid | uria | ĠPAT | rains |
| 3 | Ġ- | Ġdetermin | Ġexplos | Ġbasic | Ġcritic | Layer | ost | ĠEdgar | irc |
| | | | | *Gemma (Target Tokenizer)* | | | | | |
| 1 | Ġj | Cor | Tools | kernel | ĠLed | Ġcommittee | Ġmang | Ġcru | Ġcholesterol |
| 2 | Ġdar | Ġequality | directed | sentence | COUNT | ĠUND | ial | Ġcal | Ġmolecule |
| 3 | ba | Lex | afx | messages | Ġglycine | Ġfactors | Ġrebut | Ġmalt | apor |

- BPE algorithm (Sennrich et al., 2016) divides words into the sub-word units, also called tokens, from the statistical co-occurrence information. There may be less superficial semantic information in the tokens divided compared with words in the natural language.

- The GloVe vector for each token is obtained from the token-token co-occurrence information. These aligned tokens often appear together, e.g., 科学(science) and "Ġcritic", 왜(why) and "rains".

Therefore, it is better to choose a matric to quantify the performance of the alignment matrix learned, for example, the BLEU score in Section 2.2 or the perplexity of the initialized model.

## C  LANGUAGE CODES

We provide details of languages involved in Table 9. Following Lai et al. (2023), languages are divided by the data ratios in CommomCrawl: High (>1%), Medium (>0.1%), and Low (>0.01%).

Table 9: Details of Language codes in this work.

| ISO 639-1 | Language | Family | ISO 639-1 | Language | Family |
|---|---|---|---|---|---|
| AR | Arabic | Afro-Asiatic | TA | Tamil | Dravidian |
| BN | Bengali | Indo-European | TE | Telugu | Dravidian |
| DE | German | Indo-European | TH | Thai | Kra-Dai |
| EN | English | Indo-European | UR | Urdu | Indo-European |
| JA | Japanese | Japonic | VI | Vietnamese | Austroasiatic |
| KO | Korean | Koreanic | ZH | Chinese | Sino-Tibetan |

