# OpenReview forum: "Unifying Vocabulary of Large Language Model with Statistical Token-level Alignment"
_ICLR.cc/2025/Conference — Submitted to ICLR 2025_

### Official Review · Reviewer_8LXg · 2024-10-22

**Soundness:** 3
**Presentation:** 2
**Contribution:** 2
**Rating:** 6
**Confidence:** 4

**Summary:**

This paper wants to address the mismatch among different vocabularies used by various LLMs. UnifyVocab is proposed. The high-level idea is to use the embeddings of the tokens from the source tokenizer to initialize the embeddings of tokens from the target tokenizer. To achieve this, the authors train GloVe embeddings for the tokens in the source and target vocabularies respectively, and then align the tokens using the similarity between the source and target tokens.

**Strengths:**

- The paper is generally easy to follow.

- The experiments are extensive.

- UnifyVocab seems to be simple and effective in aligning tokens among the vocabularies of two tokenizers.

**Weaknesses:**

- The method is sensitive to the selection of the corpus used to learn the token-token alignment.

- The pipeline is very similar to WECHSEL [1]. If I understand correctly, the method proposed in this work is a simple extension to the scenario where the source and the target languages are the same (in WICHSEL they are different).

- WECHSEL additionally needs to align the learned fastText embeddings because the source and target embeddings are in different spaces. I guess this step is omitted in UnifyVocab because the authors assume the learned token GloVe embeddings (for tokenizer A and tokenizer B) are in the same space. However, this assumption might not hold true. Two embedding matrices learned from the same corpus can be quite different, even if they have the same vocabulary (and in your case, this does not hold true) [2].

- I am not sure if I agree the motivation of the paper is well-established. If a model performs well with its own tokenizer (e.g., LLama and the LLama tokenizer), why would one be interested in exchanging its tokenizer with another model's tokenizer that is intended to work on the same domain or language? I think replacing the tokenizer is mostly only meaningful when we want to have a new domain or a new language to adapt to.

[1] https://arxiv.org/abs/2112.06598
[2] https://arxiv.org/pdf/2209.15430

**Questions:**

$\textbf{Questions/Suggestions}$:

- I don't think Figure 3 (b) is meaningful. The authors claim that there is a negative relationship between the first-step training loss nad the BLEU. But the BLEU is very very bad, only around 2.4. For such a small BLEU, the differences between different initializations are basically negligible.

- In Table 4, does "0" in the column "#Tune (B)" without any training? In other words, does that line indicate the performance of right after replacing the tokenizer? If it is, maybe the authors can make it more clear in the caption.

- It is better to use the same color and same order in the legend of Figure 3 for better consistency.

- There is one related paper [3] for zero-shot tokenizer transfer. They proposed ZETT where a hypernetwork is used to predict embeddings of the new tokens in the target tokenizer. The authors may consider this as a stronger baseline method.


$\textbf{Typos}$:

Line 174: "which belongs both vocabularies." -> "which belongs to overlapping vocabularies."
Line 177: "randomly chosen token from the source vocabulary." -> "a randomly chosen token from the source vocabulary."


[3] https://arxiv.org/abs/2405.07883

---

> ### Author Response · Authors · 2024-11-28
> **Response to Reviewer 8LXg (1/2)**
>
> Thank you for your detailed review! We are thankful for your efforts during the challenging review period. We will address your concerns point by point.
>
> > The method is sensitive to the selection of the corpus used to learn the token-token alignment.
>
> To further investigate the impact of the corpus used, we replace the corpus with the SlimPajama[1] which is commonly used in the pre-training of the language model. Under 1B tokens amount, the corpus tokenized covers 98.83% of token IDs from the vocabulary of Pythia (49714/50304≈98.83%) and 97.21% of token IDs from the vocabulary of Gemma (248857/256000≈97.21%). Results are shown in the "w/ SlimPajama" row in Table 1, which reports a comparable results with the original settings. It further demonstrates the robustness of our method on the pre-training corpus for token embedding and alignment matrix.
>
> [1] Daria Soboleva, Faisal Al-Khateeb, Robert Myers, Jacob R Steeves, Joel Hestness, and Nolan Dey. SlimPajama: A 627B token cleaned and deduplicated version of RedPajama. https://huggingface.co/datasets/cerebras/SlimPajama-627B.
>
> > The pipeline is very similar to WECHSEL [1]. If I understand correctly, the method proposed in this work is a simple extension to the scenario where the source and the target languages are the same (in WICHSEL they are different).
>
> We realize that there may have been some misunderstanding regarding our method. There are two key differences between WECHSEL[2] and UnifyVocab:
>
> 1) WECHSEL needs two static **word** embedding for the source tokenizer and target tokenizer and an additional bilingual dictionary for alignment, while our method only requires a pre-training corpus to train the token-token alignment matrix. Besides, the training and aligning the GloVe embedding for each **subword/token** only cost less than 1 hour for a machine with 128 cores CPU.
>
>  2) The initialization of WECHSEL for the embedding of the language model is composed of the weighted sum of similar tokens, while UnifyVocab only re-arranges the source embedding using a one-to-one mapping function based on the token-token aligned matrix.
>
> The pipelines for adapting new vocabulary for most of the methods are similar: first, initialize the embedding for the target tokenizer and then fine-tune the initialized model. Most of the differences between methods come from the initialization of embeddings for the target vocabulary.
>
> [2] WECHSEL: Effective initialization of subword embeddings for cross-lingual transfer of monolingual language models. https://aclanthology.org/2022.naacl-main.293
>
> > WECHSEL additionally needs to align the learned fastText embeddings because the source and target embeddings are in different spaces. I guess this step is omitted in UnifyVocab because the authors assume the learned token GloVe embeddings (for tokenizer A and tokenizer B) are in the same space. However, this assumption might not hold true. Two embedding matrices learned from the same corpus can be quite different, even if they have the same vocabulary (and in your case, this does not hold true) [2].
>
> Thank you for insightful suggestions! Following Moschella et al. (2023)[3], we convert the glove embeddings into relative representations using 300 common tokens that occur in both vocabularies and conduct the left procedure of UnifyVocab to adapt the Gemma tokenizer for ${Pythia}_{1b}$. It comes to the slightly better results, which are labeled "+ Align Rep." in Table 1.
>
> [3] Relative representations enable zero-shot latent space communication. https://arxiv.org/abs/2209.15430
>
> >  I am not sure if I agree the motivation of the paper is well-established. If a model performs well with its own tokenizer (e.g., LLama and the LLama tokenizer), why would one be interested in exchanging its tokenizer with another model's tokenizer that is intended to work on the same domain or language? I think replacing the tokenizer is mostly only meaningful when we want to have a new domain or a new language to adapt to.
>
> The importance of replacing the tokenizer lies in the **fast token-level knowledge transfer from capable models**, and reducing the huge cost to train a model from scratch when a much better tokenizer is found, where **the special case is cross-lingual or cross domain vocabulary adaptation problem**. Experiments in Section 4.1 show that token-level distillation with capable language models like LLaMA3 can significantly improve the performance of Pythi $a_{1b}$ , which is comparable with vanilla ${Pythia}_{7b}$ after only 235M tokens of token-level distillation. Moreover, UnifyVocab can also be applied to the vocabulary adaptation problem for a new language or domain (Section 4.2) and achieves better results than the traditional cross-lingual vocabulary adaptation method Focus.

---

> ### Author Response · Authors · 2024-11-28
> **Response to Reviewer 8LXg (2/2)**
>
> > I don't think Figure 3 (b) is meaningful. The authors claim that there is a negative relationship between the first-step training loss nad the BLEU. But the BLEU is very very bad, only around 2.4. For such a small BLEU, the differences between different initializations are basically negligible.
>
> We argue that the low value of BLEU in Figure 3, which is the average value of  BLEU-1 to BLEU-4, comes from the low BLEU-3 and BLEU-4. It is reasonable for a one-to-one mapping alignment matrix to obtain a nearly 0 value for BLEU-3 and BLEU-4.
> Thus we provide the results of the BLEU-1 metric and an additional semantic metric named BertScore in Figure 7. Specifically, the token ID corpus $C_{t}^{'}$, which is converted by the aligned matrix from $C_s$, is de-tokenized into a text corpus $C^{'}$ by the target Tokenize$r_t$. Then we adopt "all-mpnet-base-v2" to quantify the semantic similarity between $C^{'}$ and  $C$, which is BertScore(sentence1=$C$, sentence2=$C^{'}$).
>
> As shown in Figure 7, given different alignment matrix learned, the deviation of BLEU-1 score has increased to 8~16. Moreover, we can find a similar negative relationship between the initial training loss and BLEU-1 (slope rate=-31.186, $R^2$=0.5233) or BERTScore (slope rate=-15.737, $R^2$=0.1098).
>
> > In Table 4, does "0" in the column "#Tune (B)" without any training? In other words, does that line indicate the performance of right after replacing the tokenizer? If it is, maybe the authors can make it more clear in the caption.
>
> Thanks for your helpful suggestion! As you mentioned, "#Tune (B)" denotes the performance of model after initialization without any training. We have added more descriptions in the title of Table 4.
>
> > It is better to use the same color and same order in the legend of Figure 3 for better consistency.
>
> Thanks for your suggestion! We re-arrange the order in the legend of Figure 3 as you suggested.
>
> > There is one related paper [3] for zero-shot tokenizer transfer. They proposed ZETT where a hypernetwork is used to predict embeddings of the new tokens in the target tokenizer. The authors may consider this as a stronger baseline method.
>
> Thank you for reminding us! We have supplemented the results of ZeTT[4] on ${Pythia}_{1b}$ in Table 1. Our method (97.63%) recovers more performance than the other strong baseline methods like ZeTT (91.93%) given the same amount of tokens. The hyper-network trained in ZeTT costs 418.9 GPU hours for ${Pythia}_{1b}$ on a 8*A100 80GB server, while the initialization of our method only costs less than two hours for a CPU server with 128 cores to train the GloVe embedding for token-token alignment.
>
> [4] Zero-Shot Tokenizer Transfer. https://arxiv.org/abs/2405.07883
>
> > Typos in line 174 and 177
>
> Thanks for your reminder! We have corrected the typos in the Line 174 and 177.

---

> > ### Comment · Reviewer_8LXg · 2024-11-28
> >
> > Dear authors,
> >
> > I appreciate your efforts in improving the paper's quality and adding new baselines. I am sorry that I don't have enough time to detailedly read the paper again but I looked at the most important parts. The new results seem good. The results of OFA are shown in Table 1 but not introduced in the Baseline paragraph.  The newly added method + Align Rep. should be introduced before its performance is listed. But anyway, I think the paper is better but the presentation can be further improved a bit. I increased my score to 5.
> >
> > I have some comments on your reply as follows:
> >
> > > WECHSEL needs two static word embedding for the source tokenizer and target tokenizer and an additional bilingual dictionary for alignment, while our method only requires a pre-training corpus to train the token-token alignment matrix. Besides, the training and aligning the GloVe embedding for each subword/token only cost less than 1 hour for a machine with 128 cores CPU.
> >
> > Actually, WECHSEL does not necessarily require **word** but **subword** embeddings are also OK. So I don't think this is a major difference from your method to WECHSEL.
> >
> > > The initialization of WECHSEL for the embedding of the language model is composed of the weighted sum of similar tokens, while UnifyVocab only re-arranges the source embedding using a one-to-one mapping function based on the token-token aligned matrix.
> >
> > This is true. But I think one-to-one mapping is just a simplified version of weighted sum. But to assure you, I don't think this is a weakness. I think it is just good to discuss the differences in the paper.
> >
> > > As shown in Figure 7, given different alignment matrix learned, the deviation of BLEU-1 score has increased to 8~16. Moreover, we can find a similar negative relationship between the initial training loss and BLEU-1 (slope rate=-31.186,
> > =0.5233) or BERTScore (slope rate=-15.737,
> > =0.1098).
> >
> > I guess you mean Figure 8? Yeah, that makes sense. But to be honest, I would prefer figure 8 much more than figure 3(b). As I mentioned, the very bad BLEU scores (you said the average) do not tell many things.

---

> > > ### Author Response · Authors · 2024-11-28
> > > **Response to the comment of Reviewer 8LXg**
> > >
> > > Dear Reviewer 8LXg,
> > >
> > > Thank you very much for your timely reply! It is kindly reminding that the discussion period has been extended by six days. Besides, we further improve the presentation of our paper following your helpful suggestions.
> > >
> > > > The difference between WECHSEL and UnifyVocab
> > >
> > > The method of WECHSEL to calculate the semantic representation of token t, which adds all representations of words contain the token t, is different to ours. UnifyVocab obtains the semantic representation of token t from token-token co-occurrence information in the token ID corpus. We will further compare the results of two methods in the final version.
> > >
> > > > Figure 7 --> Figure 8
> > >
> > > Yes, we forget to change the id of Figure after adding the results of convergence (Appendix B.2). Thanks for your insightful suggestions again! We will move the results of BLEU-1 into the main content in the next version.
> > >
> > > We sincerely hope that our response could resolve all of your concerns. Should there be any remaining concerns, we are more than willing to engage in further discussion to address them.
> > >
> > > If our response has satisfactorily resolved your concerns, we would be grateful if you could kindly consider providing us with a positive overall rating.
> > >
> > > Thank you in advance for your understanding.
> > >
> > > Best,
> > >
> > > The Authors of Paper 1726

---

> > > > ### Comment · Reviewer_8LXg · 2024-11-29
> > > >
> > > > Dear authors,
> > > >
> > > > Thanks for your response. My major concerns have been addressed. But as I mentioned the presentation of the paper should still be improved. E.g., make sure that you introduce/discuss your methods/baselines before you list their results.
> > > >
> > > > I increased the score to 6 since I am more positive towards the paper and good luck with the submission.

---

> > > > > ### Author Response · Authors · 2024-11-29
> > > > > **Response to the comment of Reviewer 8LXg**
> > > > >
> > > > > Dear reviewer 8LXg,
> > > > >
> > > > > Thanks again for your insightful suggestions, which benefit our paper a lot! We will improve our presentation further as you suggested.
> > > > >
> > > > > It's lucky to meet a responsible reviewer like you!
> > > > >
> > > > > Best,
> > > > >
> > > > > The authors of paper 1726

---

### Official Review · Reviewer_qU5A · 2024-11-03

**Soundness:** 3
**Presentation:** 3
**Contribution:** 3
**Rating:** 6
**Confidence:** 3

**Summary:**

This paper tackles a vocabulary extension issue in LLMs and introduce a method called UnifyVocab to replace the vocabulary of LLM, aligning token IDs between two vocabularies. The proposed approach allows vocabularies of LLMs to get replaced based on the token-token co-occurences, enabling new vocabulary adaptation with lower costs. Experimental results show some effectiveness in (cross-lingual) knowledge transfer between models.

**Strengths:**

- proposes vocabulary adaptation technique which will be useful in multilingual/crosslingual LLM  application
- Experimental results show some effectiveness of the proposed approach in multiple multilingual NLP tasks

**Weaknesses:**

- There is some missing citation on vocabulary adaptation like [1]. Comparison and/or discussion would be required.
[1] OFA: A Framework of Initializing Unseen Subword Embeddings for Efficient Large-scale Multilingual Continued Pretraining. In Proc of NAACL2024 findings

**Questions:**

- Have you ever tried other (semantic) metrics like COMET scores instead of BLEU while evaluating the performance of alignment Matrix?

---

> ### Author Response · Authors · 2024-11-28
> **Response to Reviewer qU5A**
>
> Thank you for your insightful review! We are thankful for your efforts during the challenging review period. We will address your concerns point by point.
>
> > Missing vocabulary adaptation baselines
>
> Thanks for your reminder! OFA[1] and WECHSEL[2] require additional embeddings for the source language and target language to compose the parameters of tokens in the target tokenizer. Our method reuses the parameter of the most similar source token from the aligned matrix learned by the tokenized corpus. We have supplemented the results of the following three strong baselines[1-3] on ${Pythia}_{1b}$ in Table 1. Our method (97.63%) recovers more performance than the other strong baseline methods like ZeTT (91.93%) given the same amount of tokens.
>
> [1]  OFA: A Framework of Initializing Unseen Subword Embeddings for Efficient Large-scale Multilingual Continued Pretraining.  https://aclanthology.org/2024.findings-naacl.68
>
> [2] WECHSEL: Effective initialization of subword embeddings for cross-lingual transfer of monolingual language models. https://aclanthology.org/2022.naacl-main.293
>
> [3] Zero-Shot Tokenizer Transfer. https://arxiv.org/abs/2405.07883
>
> > More semantic metrics to evaluate the performance of alignment Matrix
>
> Thanks for your suggestion! We investigate another semantic metric BertScore for COMET requires the source input. Specifically, the token ID corpus $C_{t}^{'}$, which is converted by the aligned matrix from $C_s$, is de-tokenized into a text corpus $C^{'}$ by the target Tokenize$r_t$. Then we adopt "all-mpnet-base-v2" to quantify the semantic similarity between $C^{'}$ and  $C$, which is BertScore(sentence1=$C$, sentence2=$C^{'}$). [ Test corpus $C$ --(tokenized by the source Tokenize$r_s$) → $C_s$ --(converted by align matrix $M_{s\to t}$ using the most similar target token) → $C_{t}^{'}$ --(de-tokenized by the target Tokenize$r_t$) → $C^{'}$ ]. As shown in Figure 8(b), we find that the BertScore($C$, $C^{'}$) is negatively proportional (slope rate=-15.737, $R^2$=0.1098) with the initial training loss. In other words, the semantic metric BertScore can also be used to the evaluation of alignment matrix.

---

> > ### Author Response · Authors · 2024-11-30
> > **Looking Forward to Further Discussions**
> >
> > Dear reviewer qU5A,
> >
> > We sincerely thank you for your time and effort in reviewing our paper and hope that our response could resolve all of your concerns! Should there be any remaining concerns, we are more than willing to engage in further discussion to address them.
> >
> > Best,
> >
> > The authors of paper 1726

---

> ### Author Response · Authors · 2024-12-02
> **Friendly Reminder to Review the Response**
>
> Dear Reviewer qU5A,
>
> As the discussion phase is approaching the end, we sincerely hope **you could find some time to review our response**. We hope to fully address your concerns.
>
> We understand that your time is valuable and you may be busy with other things. However, your insights would be extremely valuable for improving our work.
>
> We greatly appreciate your consideration.
>
> Best,
>
> The Authors

---

### Official Review · Reviewer_1XM4 · 2024-11-04

**Soundness:** 3
**Presentation:** 2
**Contribution:** 2
**Rating:** 5
**Confidence:** 4

**Summary:**

This paper proposes UnifyVocab, a method to replace the vocabulary of an LLM.
This involves using a tokenizer from another model and training new GloVe embeddings, which are aligned with cosine similarity to an existing embedding set, and then are used to replace the original embedding matrix and finetuned with the model.

Experiments use the Pythia base model and training corpus, and experiment with replacing the vocabulary with those from Gemma, Qwen2, and LLaMa 2 & 3.
UnifyVocab is compared to a random initialization, random permutation, and FOCUS from Dobler & de Melo (2023).
English results are compared across 6 standard tasks, and cross-lingual transfer is compared for 12 languages (+English) on 4 standard tasks.
Results show that the method preserves on average 98% of the original performance, and leads to improved cross-lingual transfer compared to FOCUS.
Two-stage tuning (first finetuning the vocabulary-related parameters in the model with the rest frozen, and then fine-tuning the full model) improves performance compared to fine-tuning the full model directly.
Token-level distillation requires less training data and generally leads to improved performance over sequence-level distillation.

The method, though, requires ~10B tokens for training, which is a significant cost compared to past approaches applied to e.g. machine translation where separately trained embeddings may be adapted to work with a model with <20k tokens.
Aligning embeddings with cosine similarity assumes that a) similar representation spaces are learned and so an explicit alignment step is not needed and b) the vocabularies are near-isomorphic, which are not guaranteed with the procedure used, and these assumptions are not mentioned. It would be easier to trust that the results would generalize if these were explored here and for example an explicit alignment step compared and more specific analysis about the conditions where the method is and is not successful (for example, if 6% vocabulary overlap with Gemma and Pythia makes the model much slower to converge, how similar is this to random initialization? are the cosine similarities considerably lower in this case, and/or less one-to-one mappings chosen? if something other than cosine similarity were used, how would this change?)

**Strengths:**

Straightforward method to replace the tokenizer / vocabulary of an LLM, given sufficient data.

**Weaknesses:**

Method is costly and does not consistently recover original model's performance. Insufficient analysis to understand the conditions where the method will succeed.

**Questions:**

Presentation note: only the best vocabulary replacement results in the tables. In Tables 4 and 5 there are times when the original Pythia model outperforms any of the replacement methods, and so should likely be bolded instead so that this is clear.

---

> ### Author Response · Authors · 2024-11-28
> **Response to Reviewer 1XM4**
>
> Thank you for your insightful review! We are thankful for your efforts during the challenging review period. We will address your concerns point by point.
>
> > Method is costly and does not consistently recover original model's performance.
>
> We realize that there may have been some misunderstanding regarding vocabulary adaptation methods for large language models.
>
> 1) **Previous works for the vocabulary adaptation of LLM cost a similar or more amount of tokens**, e.g., 65.5B for GPT-2 (WECHSEL)[1], 12.8B for XLM-R (Focus)[2]. In this work, the cost of 10B tokens mainly comes from the 2M tokens batch size, which follows the pre-training setting of Pythia, and training steps are only 5k. However, as shown in Table 6, the amount of tokens required can be reduced to 2B tokens by decreasing the batch size to 1M and training steps to 2k, which recovers the average 95.66% performance of the vanilla model.
>
> 2) **The performance of the original model is hard to recover with a limited token amount for vocabulary adaptation comparing the one of pre-training** (10B << 300B token amount in pre-training). The results of Table 2 in ZeTT[3], which replaces the Mistral tokenizer(32.0k) to the one of GPT-2 (50.3k), are further demonstrated that the phenomenon of performance loss during replacing another tokenizer. We supplement the results of the other three vocabulary adaptation methods on Pythia. As shown in Table 1, our method (97.63%) recovers more performance than the other strong baseline methods like ZeTT (91.93%) given the same amount of tokens. The hyper-network trained in ZeTT costs 418.9 GPU hours for $Pythia_{1b}$ on a 8*A100 80GB server, while the initialization of our method only requires less than two hours for a cpu server with 128 cores to train the GloVe embedding for token-token alignment.
>
> [1] WECHSEL: Effective initialization of subword embeddings for cross-lingual transfer of monolingual language models. https://aclanthology.org/2022.naacl-main.293
>
> [2] FOCUS: Effective Embedding Initialization for Monolingual Specialization of Multilingual Models. https://aclanthology.org/2023.emnlp-main.829
>
> [3] Zero-Shot Tokenizer Transfer. https://arxiv.org/abs/2405.07883
>
> > Aligning embeddings with cosine similarity assumes that a) similar representation spaces are learned and so an explicit alignment step is not needed and b) the vocabularies are near-isomorphic, which are not guaranteed with the procedure used, and these assumptions are not mentioned.
>
> Thank you for insightful review! To meet these assumptions, we follow the relative representation alignment method from Moschella et al. (2023)[4]. Specifically, we convert the GloVe embeddings into relative representations using 300 common tokens that occur in both two vocabularies, and conduct the left procedures of UnifyVocab to adapt the Gemma tokenizer for Pythia. It comes to the slightly better results, which are denoted "+ Align" in Table 1.
>
> [4] Relative representations enable zero-shot latent space communication. https://arxiv.org/abs/2209.15430
>
> > More specific analyses. For example, if 6% vocabulary overlap with Gemma and Pythia makes the model much slower to converge, how similar is this to random initialization? are the cosine similarities considerably lower in this case, and/or less one-to-one mappings chosen? if something other than cosine similarity were used, how would this change?
>
> As shown in Figure 7(a), we plot the training loss of random initialization under replacing different tokenizers, which shows a similar phenomenon in Figure 4(a) where the convergence of Gemma are slower than the one of llama3 and qwen2.
>
> We argue that the difference in converge rate may come from the different initial training losses for the one-to-one mappings chosen in our method. To evaluate this hypothesis and imitate the cases of worse methods other than cosine similarity are adopted, we randomly shuffled the learned alignment matrix for Qwen2 [40%, 60%, 80%]. The initial training loss increases from 5.35 to 11.06(randomly shuffle 80%) when replacing the Qwen2 tokenizer for $Pythia_{1b}$. As shown in Figure 7(b), the convergence of randomly shuffle 80% token-token alignment is approaching the ones of Gemma in Figure 4(a).
>
> > Presentation note in Table 4 and 5
>
> Thanks for your helpful suggestions! We have bold the better performance from vanilla model in Table 4 and 5 as suggested.

---

> > ### Author Response · Authors · 2024-11-30
> > **Looking Forward to Further Discussions**
> >
> > Dear reviewer 1XM4,
> >
> > We sincerely thank you for your time and effort in reviewing our paper and hope that our response could resolve all of your concerns! Should there be any remaining concerns, we are more than willing to engage in further discussion to address them.
> >
> > Best,
> >
> > The authors of paper 1726

---

> ### Author Response · Authors · 2024-12-02
> **Friendly Reminder to Review the Response**
>
> Dear Reviewer 1XM4,
>
> As the discussion phase is approaching the end, we sincerely hope **you could find some time to review our response**. We hope to fully address your concerns.
>
> We understand that your time is valuable and you may be busy with other things. However, your insights would be extremely valuable for improving our work.
>
> We greatly appreciate your consideration.
>
> Best,
>
> The Authors

---

### Comment · Area_Chair_H97t · 2024-11-21
**Reminder: Please respond and update the score if necessary**

Dear Reviewers,

Kindly ensure that you respond proactively to the authors' replies (once they are available) so we can foster a productive discussion. If necessary, please update your score accordingly. We greatly appreciate the time and effort you’ve dedicated to the review process, and your contributions are key to making this process run smoothly.

Thank you,

AC

---

### Public Comment · ~Xinke_Jiang1 · 2024-11-26
**Good Work but not Complete**

Hello, after reading your paper, I believe this idea is a very good piece of work, and the experiments conducted by the authors are quite thorough. However, I noticed that there are many missing details in the description, which I suspect might be due to the authors rushing to meet the deadline of ICLR. For example, the description of progressive adaptation in section 2.3 and the explanation of Figure 2 are somewhat confusing to the readers. It would be great if the authors could elaborate on these aspects to improve the clarity of the work.

---

> ### Author Response · Authors · 2024-11-28
> **Response to public comment**
>
> Thank you for your appreciation to our work! We supplement more descriptions, experiments and analyses to support the claims and effectiveness of our method. If you have any other confusion, please feel free to raise more comments :)

---

### Author Response · Authors · 2024-11-28
**Rebuttal Revision by Authors**

- Following the suggestions of Reviewer qU5A and 8LXg, we add the results of three strong baseline methods for vocabulary adaptation.[1-3]
- Following the suggestions of Reviewer 1XM4 and 8LXg, we add the GloVe embedding alignment process using the relative representation method from Moschella et al. (2023) [4] to meet the assumption of aligning embeddings with cosine similarity. Results are reported in Table 1.
- Following the suggestions of Reviewer 1XM4, we further investigate the convergence of random initialization and the impact of different alignment matrices learned to the training convergence (Appendix B.3).
- We supplement the results of other two metrics, BLEU-1 and BertScore, to evaluate the performance of alignment matrix learned (Appendix B.4).

[1] WECHSEL: Effective initialization of subword embeddings for cross-lingual transfer of monolingual language models. https://aclanthology.org/2022.naacl-main.293

[2] FOCUS: Effective Embedding Initialization for Monolingual Specialization of Multilingual Models. https://aclanthology.org/2023.emnlp-main.829

[3] Zero-Shot Tokenizer Transfer. https://arxiv.org/abs/2405.07883

[4] Relative representations enable zero-shot latent space communication. https://arxiv.org/abs/2209.15430

---

### Meta-Review · Area_Chair_H97t · 2024-12-22

**Metareview:**

The paper proposes UnifyVocab, a method to replace and harmonize vocabularies in large language models (LLMs). It uses new GloVe embeddings aligned with existing ones via cosine similarity, replacing original embeddings and allowing the model to be fine-tuned. Experiments with various vocabularies show that UnifyVocab maintains 98% of original performance while improving cross-lingual transfer compared to other methods. The two-stage tuning further enhances performance, although the approach is resource-intensive, requiring about 10 billion tokens for training. The paper highlights potential improvements in embedding alignment assumptions and suggests that UnifyVocab effectively addresses vocabulary mismatches across LLMs, enabling cost-effective adaptation and improved cross-lingual knowledge transfer.

This paper received borderline scores. While the idea has merit, and reviewers agree that the method could be valuable for token embedding initialization, the overall presentation requires substantial improvement. The proposed method also has notable drawbacks, particularly concerning its efficiency and its similarities to prior work. Additional issues include unclear and poorly presented results, especially in Table 4, which reports perplexity and needs significant refinement. Moreover, the paper lacks strong motivation. Although the authors attempted to address this in their response, the paper itself needs to provide a clearer and more compelling rationale. Given the issues outlined above, I am leaning towards recommending the rejection of the paper.

**Additional Comments On Reviewer Discussion:**

Reviewer 8LXg expressed concerns regarding the similarities between the proposed work and existing research, as well as the method's efficiency. However, following the discussion, the reviewer was satisfied with the response provided by the authors, who included a comparison between WECHSEL and UnifyVocab during the author response period. Additionally, the reviewer noted that the paper lacks a strong motivation, raising the question of why this method is necessary for transfer learning. Reviewer 1XM4 further emphasized the issue of insufficient analysis within the paper. Reviewer 1XM4 also highlighted the presentation issue.

---

### Decision · Program_Chairs · 2025-01-22

Reject